# Diffusion-Weighted Magnetic Resonance Imaging for the Diagnosis of Lymph Node Metastasis in Patients with Biliary Tract Cancer

**DOI:** 10.3390/cancers16183143

**Published:** 2024-09-13

**Authors:** Takashi Murakami, Hiroaki Shimizu, Hiroyuki Nojima, Kiyohiko Shuto, Akihiro Usui, Chihiro Kosugi, Keiji Koda

**Affiliations:** Department of Surgery, Teikyo University Chiba Medical Center, Ichihara 299-0112, Japan; gtrennsport3@gmail.com (T.M.); kshuto@med.teikyo-u.ac.jp (K.S.); ausui@med.teikyo-u.ac.jp (A.U.); ckosugi0126@yahoo.co.jp (C.K.); k-koda@med.teikyo-u.ac.jp (K.K.)

**Keywords:** diffusion-weighted imaging, magnetic resonance imaging, lymph node metastasis, biliary tract cancer, diagnostic accuracy

## Abstract

**Simple Summary:**

This study assessed the preoperative diagnostic efficacy of the apparent diffusion coefficient on diffusion-weighted magnetic resonance imaging for identifying lymph node metastasis in biliary tract cancer, comparing it with the diagnostic efficacy of short-axis and long-axis diameters of lymph nodes measured by computed tomography. The results demonstrated that the minimum ADC value provided the highest diagnostic accuracy.

**Abstract:**

**Objective**: The diagnostic efficacy of the apparent diffusion coefficient (ADC) in diffusion-weighted magnetic resonance imaging (DW-MRI) for lymph node metastasis in biliary tract cancer was investigated in the present study. **Methods**: In total, 112 surgically resected lymph nodes from 35 biliary tract cancer patients were examined in this study. The mean and minimum ADC values of the lymph nodes as well as the long-axis and short-axis diameters of the lymph nodes were assessed by computed tomography (CT). The relationship between these parameters and the presence of histological lymph node metastasis was evaluated. **Results**: Histological lymph node metastasis was detected in 31 (27.7%) out of 112 lymph nodes. Metastatic lymph nodes had a significantly larger short-axis diameter compared with non-metastatic lymph nodes (*p* = 0.002), but the long-axis diameter was not significantly different between metastatic and non-metastatic lymph nodes. The mean and minimum ADC values for metastatic lymph nodes were significantly reduced compared with those for non-metastatic lymph nodes (*p* < 0.001 for both). However, the minimum ADC value showed the highest accuracy for the diagnosis of histological lymph node metastasis, with an area under the curve of 0.877, sensitivity of 87.1%, specificity of 82.7%, and accuracy of 83.9%. **Conclusions**: The minimum ADC value in DW-MRI is highly effective for the diagnosis of lymph node metastasis in biliary tract cancer. Accurate preoperative diagnosis of lymph node metastasis in biliary tract cancer should enable the establishment of more appropriate treatment strategies.

## 1. Introduction

Biliary tract cancer includes intrahepatic cholangiocarcinoma, perihilar cholangiocarcinoma, distal cholangiocarcinoma, gallbladder cancer, and ampullary cancer [1]. Biliary tract cancer is a rare malignancy, with an incidence of approximately 6 per 100,000 people, but the overall incidence of biliary tract cancer is increasing due to the increase in intrahepatic cholangiocarcinoma [2]. Computed tomography (CT) and magnetic resonance imaging (MRI) are effective for assessing the primary lesion, the extent of disease, and vascular invasion [1]. Endoscopic retrograde cholangiopancreatography (ERCP) is useful not only for diagnosing the horizontal extent of cholangiocarcinoma but also for vertical extent diagnosis with intraductal ultrasonography, as well as for pathological diagnosis through cytology or bile duct biopsy [1]. ERCP also allows for biliary drainage in cases of obstructive jaundice. Endoscopic ultrasonography has high diagnostic capability for qualitative diagnosis and vascular invasion [1]. Positron emission tomography (PET) is useful for detecting metastases and diagnosing postoperative recurrence [1].

Surgical resection is the only curative treatment for biliary tract cancer, but various factors can lead to unresectability. Surgical treatment is highly invasive and often involves concomitant liver resection; so, the patient’s overall condition and factors such as decreased liver function may result in a determination of unresectability [3]. Biliary tract cancer with distant metastasis is generally considered unresectable. However, there have been cases where it has been finally treated by conversion surgery after long-term chemotherapy [4].

The prognosis of biliary tract cancer remains poor among gastrointestinal cancers, with a 5-year survival rate of 24–61% [5]. Lymph node metastasis has been shown to be an independent prognostic factor in biliary tract cancer, including extrahepatic bile duct cancer, gallbladder cancer, and ampullary cancer [6,7,8]. Furthermore, in cases of distal cholangiocarcinoma, a good prognosis can be expected with curative resection when the number of metastatic lymph nodes is two or fewer [9]. In gallbladder cancer, the number of metastatic lymph nodes is considered a poor prognostic factor [7]. For ampullary cancer, cases with four or more metastatic lymph nodes have been reported to have a poorer prognosis than cases with three or fewer [10]. Matsuyama et al. demonstrated that lymph node metastasis and vascular invasion are poor prognostic factors in perihilar cholangiocarcinoma [8]. In a subsequent report, they described administering neoadjuvant chemotherapy with a combination of gemcitabine and S-1, a combined drug comprising tegafur, gimestat, and otastat potassium, to perihilar cholangiocarcinoma patients with risks including lymph node metastasis, with favorable outcomes including a disease control rate of 91.3%, a median survival time of 30.3 months, and a 5-year survival rate of 30%. Additionally, the resection rate in the cohort was 71%, and among the resected cases, the R0 resection rate was 81% [11]. Therefore, accurate diagnosis of preoperative lymph node metastasis is essential for biliary tract cancer.

Preoperative diagnosis of lymph node metastasis is usually performed by CT, MRI, or PET [12]. Previous findings indicated that metastatic lymph nodes had a larger long-axis diameter compared with non-metastatic lymph nodes, and the long-axis diameter correlated with the area of cancer within the lymph nodes [13]. However, diagnosis using the long-axis diameter of the lymph nodes was considered to provide low accuracy for identifying metastatic lymph nodes. Therefore, in various types of cancer, the short-axis diameter of lymph nodes was previously utilized for diagnosing lymph node metastasis [9,14]. However, when evaluating only the short-axis diameter of the lymph nodes, lymph node swelling due to inflammation may be misdiagnosed as lymph node metastasis. Noji et al. concluded that CT diagnosis is no longer useful for determining lymph node metastasis in biliary tract cancer [15].

Proliferated tumor cells increase cellular density and suppress the diffusion of water molecules, leading to higher signal intensity in diffusion-weighted MRI (DW-MRI) [16]. The apparent diffusion coefficient (ADC) value quantitatively represents this magnitude of diffusion, which can be applied to differentiate malignant tumors from normal tissues [16]. The ADC value in pancreatic cancer has been reported to be associated with the degree of tumor differentiation and identified as an independent prognostic factor [17]. This result indicates that the ADC value reflects tumor malignancy. Moreover, DW-MRI has been reported to be useful for distinguishing between malignant and benign biliary tract diseases. In the differentiation of gallbladder cancer and cholecystitis, the ADC value was found to be significantly lower in cases of gallbladder cancer [18,19]. Similarly, DW-MRI was effective for differentiating bile duct cancer from benign biliary stricture [20]. More recently, Miyazaki et al. reported that a low minimum ADC value in the primary lesion of intrahepatic cholangiocarcinoma was significantly correlated with low tumor-infiltrating lymphocytes and was shown to be an independent poor prognostic factor, suggesting the usefulness of the minimum ADC value [21]. In the present study, we investigated the diagnostic performance of DW-MRI for lymph node metastasis in biliary tract cancer.

## 2. Materials and Methods

### 2.1. Patients

Biliary tract cancer patients who underwent surgical resection at our hospital between April 2017 and April 2021 were included in this study. These patients underwent preoperative abdominal contrast-enhanced dynamic CT (arterial phase, portal phase, equilibrium phase) and MRI examinations. The surgically resected specimens were fixed in 10% formalin, embedded in paraffin, and then pathologically evaluated using hematoxylin and eosin staining. The dissected lymph nodes were examined pathologically for metastasis. The study was carried out following the Helsinki Declaration and received approval from the institutional review board at the Teikyo University Chiba Medical Center (ethical committee approval No.18-171).

### 2.2. Diagnosis of Lymph Node Metastasis

Regional and para-aortic lymph nodes detected by both CT and MRI and surgically removed by lymph node dissection were selected as subjects for the present study. The short-axis and long-axis diameters of these lymph nodes were measured by CT (Figure 1a). MRI was performed on a 1.5 T body scanner equipped with a phased array body coil (Signa HDxt 1.5 T; GE Healthcare, Chicago, IL, USA). DW-MRI was obtained using a single-shot spin-echo type of echo-planar sequence, with fat signal suppression applied through short-tau inversion recovery. The b values corresponding to diffusion-sensitizing gradients were 0 and 1000 s/mm^2^. Sequential sampling of the k-space was used with an effective echo time (TE) and an acquisition matrix of 92 × 192, which was interpolated to 256 × 256 during image calculation. Repetition time (TR) and TE were 12,857 ms and 73 ms, respectively. Slices covering the upper abdomen were acquired with a 400-mm field of view, a slice thickness of 5 mm, and a 1-mm gap between slices. T2-weighted images were obtained using the following parameters: TR/TE 618/89, a matrix of 384 × 224, a 400-mm field of view, and a 5-mm section thickness. The mean and minimum ADC values of each lymph node were measured. For quantitative analysis, each annotated lymph node was identified on the corresponding DWI; a region of interest (ROI) for each lymph node was drawn on the ADC map. Circular or oval ROIs were applied to include almost all of the visible lymph nodes; the mean of the ROIs was defined as the mean ADC value and the minimum of the ROIs as the ADC minimum value.

### 2.3. Statistical Analysis

The association between these parameters and histological lymph node metastasis was statistically evaluated using SPSS Statistics version 21.0 (IBM, New York City, NY, USA). Continuous variables are presented as means and standard deviations. The Mann–Whitney U test was used for continuous variables. The receiver operating characteristic curve was used to assess the diagnostic performance, and cutoff values were determined by receiver operating characteristic (ROC) curves. The area under the curve, sensitivity, specificity, and accuracy were calculated. A *p* value less than 0.05 was considered statistically significant.

## 3. Results

A total of 35 biliary tract cancer patients were eligible for this study, comprising 21 males and 14 females (Table 1). The mean age was 73.3 years. The biliary tract cancers in this study included perihilar cholangiocarcinoma (n = 10), intrahepatic cholangiocarcinoma (n = 1), distal cholangiocarcinoma (n = 10), gallbladder carcinoma (n = 10), and ampullary carcinoma (n = 4). The breakdown of surgical procedures included right hemihepatectomy and caudate lobectomy with extrahepatic bile duct resection in 6 cases, left hemihepatectomy and caudate lobectomy with extrahepatic bile duct resection in 6 cases, hepatectomy of segment 4a+5 with extrahepatic bile duct resection in 8 cases, hepatic segmentectomy with extrahepatic bile duct resection in 1 case, and pancreaticoduodenectomy in 14 cases. Eleven (31.4%) patients experienced preoperative cholangitis.

In total, 31 out of 112 (27.7%) lymph nodes were positive for histological metastases. The distribution of lymph nodes was as follows: common hepatic artery (n = 23), hepatoduodenal ligament (n = 58), around the pancreatic head (n = 20), jejunal mesentery (n = 4), para-aorta (n = 6), and diaphragm (n = 1) (Table 2).

Examples of CT and MRI findings for identifying lymph node metastasis before surgery in a patient with biliary tract cancer are shown (Figure 1).

The short-axis diameter of metastatic lymph nodes (6.7 ± 2.7 mm) was markedly greater than that of non-metastatic lymph nodes (5.1 ± 2.1 mm, *p* = 0.002, Figure 2a). In contrast, there was no significant difference in long-axis diameter between metastatic and non-metastatic lymph nodes (*p* = 0.99, Figure 2b). The mean ADC mean value was significantly reduced in metastatic lymph nodes (1.26 ± 0.20 × 10^−3^ mm^2^/s) compared with non-metastatic lymph nodes (1.51 ± 0.23 × 10^−3^ mm^2^/s, *p* < 0.001, Figure 2c). Moreover, the minimum ADC value was notably decreased in metastatic lymph nodes (1.01 ± 0.16 × 10^−3^ mm^2^/s) compared with lymph nodes without metastasis (1.33 ± 0.23 × 10^−3^ mm^2^/s, *p* < 0.001, Figure 2d).

The areas under the curve for the short-axis diameter, long-axis diameter, mean ADC, and minimum ADC were 0.69, 0.50, 0.80, and 0.88, respectively (Figure 3).

The cutoff values calculated from the ROC curves were as follows: short-axis diameter, 5.8 mm; mean ADC value, 1.345 × 10^−3^ mm^2^/s; and minimum ADC value, 1.140 × 10^−3^ mm^2^/s. Using these cutoff values, the sensitivity, specificity, and accuracy for differentiating histological lymph node metastasis were 58.1%, 75.3%, and 70.5% for short-axis diameter, 80.6%, 79.0%, and 79.5% for the mean ADC, and 87.1%, 82.7%, and 83.9% for the minimum ADC, respectively (Table 3).

## 4. Discussion

The present study demonstrated that the minimum ADC value within lymph nodes according to DW-MRI had the highest accuracy in diagnosing preoperative lymph node metastasis in biliary tract cancer.

The formation of lymph node metastasis is involved in tumor-associated lymphangiogenesis [22,23,24]. It has been suggested that lymphangiogenesis occurs even before the formation of lymph node metastasis, forming a pre-metastatic niche. Tumor cells undergo epithelial-mesenchymal transition, increasing their motility and invasiveness, thereby promoting invasion into the lymphatic vessels [25,26]. Tumor cells first metastasize to the draining lymph node (sentinel lymph node). Moreover, once tumor cells metastasize to the lymph nodes, lymphangiogenesis is further promoted [27]. It has been reported that large lymph node metastases can obstruct lymphatic drainage vessels, increasing intranodal pressure; thus, lymphatic flow bypasses from the sentinel lymph node to other lymph nodes [28]. Additionally, a route for metastasis from lymph nodes to other organs has also been reported [29]. In head and neck cancers, a correlation has been observed between intratumoral lymphatic vessel density and lymph node metastasis [30]. Furthermore, overexpression of vascular endothelial growth factor (VEGF)-A, VEGF-C, or VEGF-D has been shown to promote the growth of tumor-associated lymphatic vessels and to facilitate lymph node metastasis [31]. The correlation between the expression of VEGF-C or VEGF-D and lymph node metastasis has also been demonstrated in colorectal cancer, gastric cancer, and esophageal cancer, as well as in breast cancer, lung cancer, and uterine cancer [32].

Previous reports on the diagnosis of lymph node metastasis in biliary tract cancer using DW-MRI include the following. In 2016, the ADC mean value was reported to be useful for diagnosing lymph node metastasis in cholangiocarcinoma [32]. Additionally, recent studies have shown that the ADC mean value is diagnostically useful for lymph node metastasis in pancreato-biliary cancer and perihilar cholangiocarcinoma [33,34,35]. In contrast, Promsorn et al. reported that the ADC mean value did not contribute to the diagnosis of lymph node metastasis in cholangiocarcinoma [16]. They concluded that the mean ADC value was not very useful in distinguishing metastatic from nonmetastatic lymph nodes when the cancerous lesion was confined to a small portion of the lymph node. Theoretically, when the viable cancerous lesion within the metastatic lymph node occupies only a small portion of the lymph node, the mean ADC value of the lymph node is estimated to be lower, but the minimum ADC value remains unchanged. Therefore, the minimum ADC value can more accurately reflect the presence of lymph node metastasis, even if it is a small portion of the lymph node (Figure 1c,d). This may explain why the ADC minimum was even more diagnostically accurate than the ADC mean in the present study. Furthermore, abdominal CT is considered less likely to detect such micrometastases within lymph nodes compared with MRI, because the lymph nodes with micrometastases may not be enlarged.

PET has been reported to be more effective than CT in identifying lymph node metastasis in biliary tract cancer [36]. Regarding the comparison between PET and DW-MRI for diagnosing lymph node metastasis, DW-MRI is associated with higher sensitivity than PET in esophageal cancer [37]. On the other hand, combining MRI and PET information with CT can improve diagnostic accuracy in relation to lymph node metastasis [38]. Recent reports have demonstrated that CT radiomics achieved high diagnostic accuracy for lymph node metastasis in perihilar cholangiocarcinoma and intrahepatic cholangiocarcinoma [39,40]. Furthermore, the integration of artificial intelligence with radiomics has further enhanced diagnostic accuracy [41,42,43,44,45]. In rectal cancer, a model predicting lymph node metastasis by using machine learning software to perform deep learning in relation to clinicopathological factors such as age, sex, tumor markers, T factor, and short-axis diameter of lymph nodes was proven to be more useful than conventional diagnostic criteria [44]. In addition, the efficacy of MRI radiomics in diagnosing lymph node metastasis in pancreatic cancer and rectal cancer has been reported [46,47].

Incorporating these advanced technologies with DW-MRI is anticipated to enhance the accuracy of preoperative diagnosis of lymph node metastasis in biliary tract cancer. Accurate preoperative diagnosis of lymph node metastasis in biliary tract cancer would allow the establishment of more appropriate treatment strategies. For example, neoadjuvant chemotherapy could be performed in cases with positive lymph node metastasis, or resection could be avoided in cases with metastasis to extra-regional lymph nodes.

This study has several limitations. First, this study is a retrospective study with a limited sample size. To derive generalizable results, larger prospective studies involving more extensive cohorts are needed in the future. Secondly, this study used specific MRI equipment and protocols, which means that further verification is required to assess reproducibility across different MRI devices. Furthermore, the pathological diagnosis of lymph node metastasis using HE staining may miss micrometastases. Yamamoto H reported that when a one-step nucleic acid amplification assay was performed on colorectal cancer cases, the upstaging rates were 2.0% for Stage I, 17.7% for Stage IIA, 12.5% for Stage IIB, and 25% for Stage IIC [48]. However, the minimum ADC value may have the potential to detect such micrometastases accurately, and further studies should be conducted to evaluate the diagnostic capability of the minimum ADC value for lymph node micrometastasis.

## 5. Conclusions

The present study demonstrated that the short-axis diameter, mean ADC value, and minimum ADC value of the lymph nodes were significantly different between histologically metastatic and non-metastatic lymph nodes. However, the minimum ADC value showed the best discriminative ability, with the highest sensitivity, specificity, and accuracy. These findings suggest that the minimum ADC value is highly effective for differentiating metastatic lymph nodes in biliary tract cancer.

## Figures and Tables

**Figure 1 cancers-16-03143-f001:**
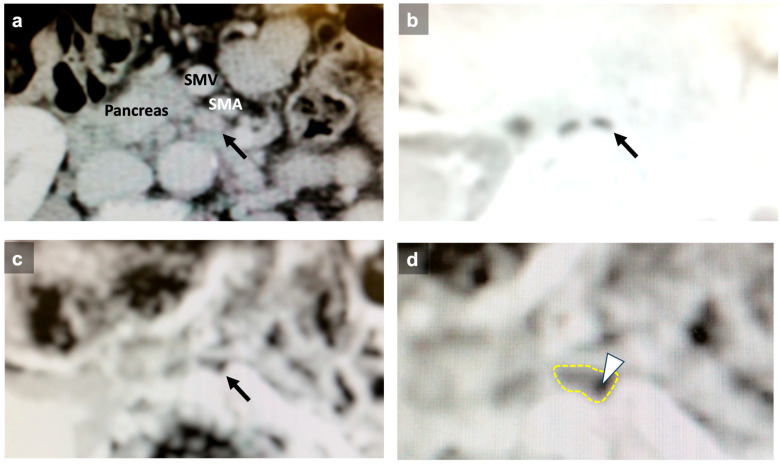
CT and MRI findings for the diagnosis of lymph node metastasis. Abdominal CT revealed an enlarged lymph node with a 6.7 mm short-axis diameter (**a**, arrow). In the DW-MRI, the lymph node showed restricted diffusion (**b**, arrow). On the ADC map, the lymph node appeared gray to black (**c**, arrow). (**d**) is a magnified image of (**c**). The mean ADC value was calculated as the average of the ROI indicated by the yellow dotted line, and the minimum ADC value reflected the area with the lowest ADC value, as shown by the arrowhead. The mean ADC value was 1.08 × 10^−3^ mm^2^/s and the minimum ADC value was 0.99 × 10^−3^ mm^2^/s. This lymph node was proven to be metastatic by histological examination. SMA, superior mesenteric artery; SMV, superior mesenteric vein.

**Figure 2 cancers-16-03143-f002:**
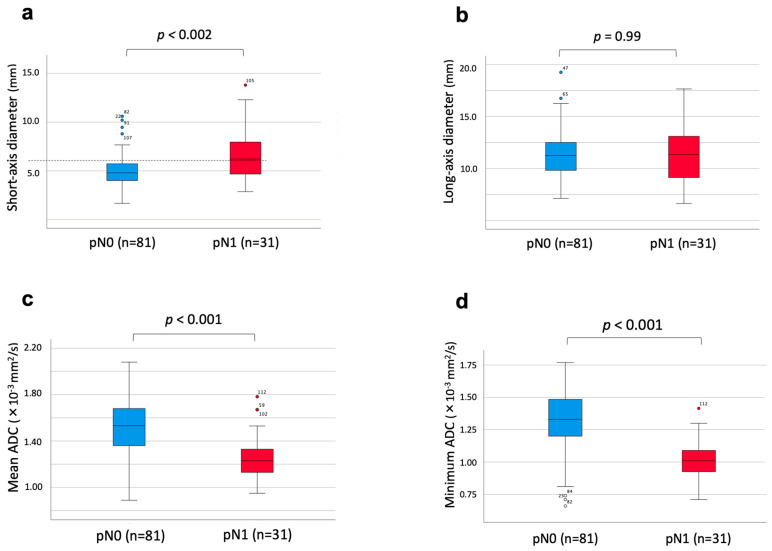
Correlation between lymph node diameters, ADC values, and histological lymph node metastasis. The short-axis diameter was significantly larger in the lymph node metastasis group (**a**), while the long-axis diameter did not correlate with the presence of lymph node metastasis (**b**). The mean ADC value was significantly lower in metastatic lymph nodes (**c**) and the minimum ADC value was significantly lower in lymph nodes without metastasis (**d**).

**Figure 3 cancers-16-03143-f003:**
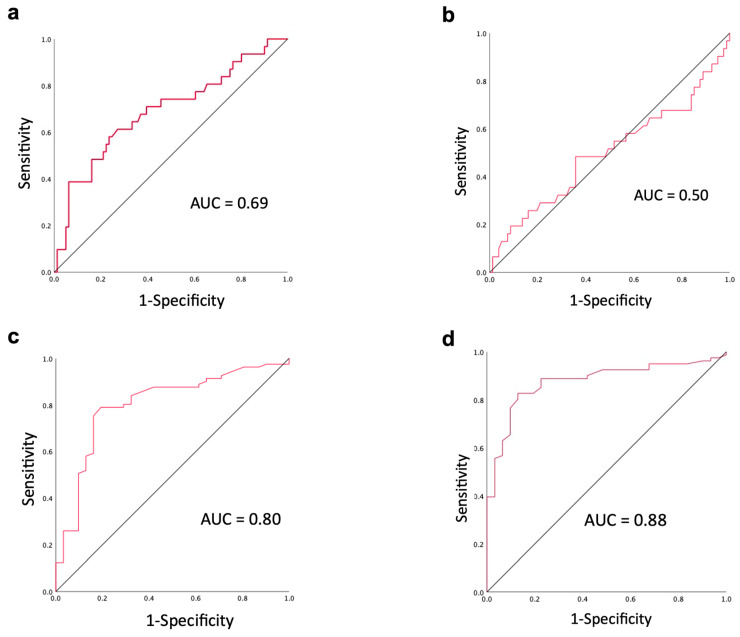
ROC curve for each diagnostic criterion. ROC curves for the short-axis diameter (**a**), long-axis diameter (**b**), mean ADC value (**c**), and minimum ADC value (**d**).

**Table 1 cancers-16-03143-t001:** Patient characteristics (n = 35).

Factors	
Age, years (mean ± SD)	73 ± 8
Sex	
Male	21
Female	14
Anatomical tumor location	
Intrahepatic cholangiocarcinoma	1
Perihilar cholangiocarcinoma	10
Distal bile duct cancer	10
Gallbladder cancer	10
Ampullary carcinoma	4
Operative procedure	
Right hemihepatectomy + caudate lobectomy + BDR	6
Left hemihepatectomy + caudate lobectomy + BDR	6
Hepatectomy of segment 4a+5 + BDR	8
Hepatic segmentectomy + BDR	1
Pancreatoduodenectomy	14
Preoperative cholangitis	
Yes	11
No	24
Lymph nodes analyzed	112
pN+	31
pN−	81

SD, standard deviation; BDR, bile duct resection; pN+; pathologically positive lymph node metastasis, pN−; pathologically negative lymph node metastasis.

**Table 2 cancers-16-03143-t002:** Details of lymph node locations.

Locations of Lymph Nodes	pN+	pN−	Total
Common hepatic artery	4	19	23
Hepatoduodenal ligament	9	49	58
Pancreatic head	12	8	20
Jejunal mesentery	2	2	4
Para-aorta	3	3	6
Diaphragm	1	0	1
Total	31	81	112

**Table 3 cancers-16-03143-t003:** Comparison of diagnostic performance.

Diagnostic Criteria for Lymph Node Metastasis	Sensitivity	Specificity	Accuracy
Short-axis diameter > 5.785 mm	58.1%	75.3%	70.5%
Mean ADC < 1.345 × 10^−3^ mm^2^/s	80.6%	79.0%	79.5%
Minimum ADC < 1.14 × 10^−3^ mm^2^/s	87.1%	82.7%	83.9%

ADC, apparent diffusion coefficient.

## Data Availability

The datasets generated and/or analyzed during the current study are available from the corresponding author on reasonable request.

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
