# Peer review of "Diffusion-Weighted Magnetic Resonance Imaging for the Diagnosis of Lymph Node Metastasis in Patients with Biliary Tract Cancer"

_cancers, 2024, doi:10.3390/cancers16183143_

Round 1

Reviewer 1 Report

Comments and Suggestions for Authors

Dear authors,

Thank you very much for your meticulous research work. The value of extended MRI diagnostics using ADC to assess the lymph node status is very well described.

I would be pleased if you could briefly outline the possible implications. Will you be subjecting more patients to neoadj. therapy in future? Will there be any changes to your indications?

Best regards and many thanks,

Reviewer 2 Report

Comments and Suggestions for Authors

1) There is a high similarity index with many parts similar to previously published material. Needs rephrasing to avoid plagiarism. 

2) Outline the significance of the objective in your abstract. 

3) Many sentences in the introduction are not supported by references, please check. 

4) Please specify the ethical approval code - Line 104.

5) Figure 1 has been cited in the methods. I believe this should be cited and explained in the results. 

6) Check the caption of the figures. Do not list (a), (b), (c), etc... as a list. It should be a continuous statement. 
